# Volume, but Not the Location of Necrosis, Is Associated with Worse Outcomes in Acute Pancreatitis: A Prospective Study

**DOI:** 10.3390/medicina58050645

**Published:** 2022-05-08

**Authors:** Inga Dekeryte, Kristina Zviniene, Edita Bieliuniene, Zilvinas Dambrauskas, Povilas Ignatavicius

**Affiliations:** 1Department of Surgery, Lithuanian University of Health Sciences, 44307 Kaunas, Lithuania; ingadeke0918@kmu.lt (I.D.); zilvinas.dambrauskas@lsmuni.lt (Z.D.); 2Department of Radiology, Lithuanian University of Health Sciences, 44307 Kaunas, Lithuania; kristinazviniene@lsmuni.lt (K.Z.); edita.bieliuniene@lsmuni.lt (E.B.)

**Keywords:** acute pancreatitis, necrotizing pancreatitis, outcomes, computed tomography, management, surgery

## Abstract

*Background and Objectives*: The course and clinical outcomes of acute pancreatitis (AP) are highly variable. Up to 20% of patients develop pancreatic necrosis. Extent and location of it might affect the clinical course and management. The aim was to determine the clinical relevance of the extent and location of pancreatic necrosis in patients with AP. *Materials and Methods*: A cohort of patients with necrotizing AP was collected from 2012 to 2018 at the Hospital of Lithuanian University of Health Sciences. Patients were allocated to subgroups according to the location (entire pancreas, left and right sides of pancreas) and extent (<30%, 30–50%, >50%) of pancreatic necrosis. Patients were reviewed for demographic features, number of performed surgical interventions, local and systemic complications, hospital stay and mortality rate. All contrast enhanced computed tomography (CECT) scans were evaluated by at least two experienced abdominal radiologists. All patients were treated according to the standard treatment protocol based on current international guidelines. *Results*: The study included 83 patients (75.9% males (*n* = 63)) with a mean age of 53 ± 1.7. The volume of pancreatic necrosis exceeded 50% in half of the patients (*n* = 42, 51%). Positive blood culture (*n* = 14 (87.5%)), multiple organ dysfunction syndrome (*n* = 17 (73.9%)) and incidences of respiratory failure (*n* = 19 (73.1%)) were significantly more often diagnosed in patients with pancreatic necrosis exceeding 50% (*p* < 0.05). Patients with >50% of necrosis were significantly (*p* < 0.05) more often diagnosed with moderately severe (*n* = 24 (41.4%)) and severe (*n* = 18 (72%)) AP. The number of surgical interventions (*n* = 18 (72%)) and ultrasound-guided interventions (*n* = 26 (65%)) was also significantly higher. In patients with whole-pancreas necrosis, incidence of renal insufficiency (*n* = 11 (64.7%)) and infected pancreatic necrosis (*n* = 19 (57.6%)) was significantly higher (*p* < 0.05). *Conclusions*: The clinical course and outcome were worse in the case of pancreatic necrosis exceeding 50%, rendering the need for longer and more complex treatment.

## 1. Introduction

Acute pancreatitis (AP) has a wide spectrum of severity, from a mild spontaneously resolving to a severe and fatal disease. Approximately 20% of patients develop necrotizing pancreatitis defined by necrosis of the parenchyma or extrapancreatic fat tissue [1]. The major causes of death are the infection of necrotic tissue and organ failure, which are associated with a poor prognosis [2]. Mortality reaches approximately 15% in patients with necrotizing AP and up to 30–39% in those with infected necrosis. Infected pancreatic necrosis is diagnosed in about one third of patients with necrosis [3,4,5]. Contrast-enhanced computed tomography (CECT) is the standard imaging modality in the setting of AP [3]. Usually, initial CECT is not routinely required for AP diagnosis unless there are diagnostic uncertainties, failure to respond to conservative treatment or need for severity confirmation. The optimal timing for the first CECT scan is 72–96 h after the onset of AP symptoms [6]. Computed tomography is used for evaluation of local complications, such as peripancreatic fluid collection and peripancreatic necrosis (sterile or infected), pseudocyst and walled-off necrosis [7]. Early and accurate prediction of disease severity is one of the first steps when choosing the optimal treatment [8]. Identification of the location of pancreatic necrosis could help predict specific complications, such as fluid collections [9]. Patients with AP are classified according to Revised Atlanta Classification or Determinant-Based Classification. Several scoring systems (BISAP, MODS and other) and single indicators such as hematocrit [6], creatinine, lactate dehydrogenase and oxygenation index [10] are used to predict the severity and outcomes of the disease. However, the clinical value of the location and extent of pancreatic necrosis as shown by CECT in prognostication of the course and outcomes in early AP remains controversial.

The aim of this study was to determine the clinical significance of the extent and location of pancreatic necrosis in predicting the course and outcomes of the disease among patients with AP.

## 2. Materials and Methods

This was nonrandomized, single-center cohort study. Data collection was performed at the Department of Surgery, Lithuanian University of Health Sciences, using a specially developed and maintained database from 2012 to 2018. Medical records were reviewed for the following data: demographic features, AP diagnostic criteria, etiology, laboratory tests, CECT timing and results, antibacterial treatment, surgical interventions, complications, American Society of Anesthesiologists (ASA) physical status class, length of hospital stay, mortality rate. The Kaunas Regional Biomedical Research Ethics Committee approved the study (protocol No. BEC-MF-232) and allowed the use of publicly unavailable databases.

### 2.1. Patients

We included 83 consecutive patients with a diagnosis of necrotizing AP. The diagnosis was based on the International Association of Pancreatology (IAP)/American Pancreatic Association (APA) evidence-based guidelines for the management of acute pancreatitis. At least two of three diagnostic criteria must have been fulfilled: (1) upper abdominal pain; (2) serum amylase >3 times the upper limit of normal; (3) ultrasonography/CT signs of AP. Location and extent of necrosis were identified by CECT. Inclusion criteria were as follows: (1) patients older than 18; (2) patients diagnosed with necrotizing AP; (3) performed CECT. Patients with a diagnosis or radiological signs of pancreatic cancer and chronic pancreatitis were excluded from this study.

Etiology of the disease was determined based on the history and available clinical data. Confirmed alcohol intake and no laboratory and/or radiological evidence of biliary obstruction were the diagnostic criteria for alcohol-induced AP. No history of alcohol intake and laboratory results suggesting biliary obstruction and/or radiological findings of biliary stones were the diagnostic criteria for biliary AP. If there was not enough evidence of alcohol-induced or biliary pancreatitis, the patient was diagnosed with idiopathic AP. Acute pancreatitis was determined as recurrent if the patient was treated for AP before. Infected necrosis was diagnosed when culture samples from peripancreatic collection were positive or there were radiological signs (i.e., gas in peripancreatic collection) of infection in an abdominal CECT scan.

Patients were allocated to subgroups according to CECT scan findings. According to the site of pancreatic necrosis: (1) entire pancreas; (2) left part of the pancreas; (3) right part of the pancreas, and according to the extent of pancreatic necrosis: (1) <30%; (2) 30–50%; (3) >50%. The extent of pancreatic necrosis was calculated by excluding peripancreatic necrosis. Then, the images were reviewed in all 3 planes (axial, coronal and sagittal), and necrosis extent of the whole pancreas was assigned to one of 3 groups (<30%, 30–50% or >50%). All unclear cases were discussed among two radiologists.

The severity of the disease was assessed using Revised Atlanta and Determinant-based classifications. Multiple organ dysfunction score (MODS) was assessed on the day of admission and on the day of sepsis manifestation if it was diagnosed.

### 2.2. Computed Tomography

In all cases, CT scans were performed later than 72 h from the onset of AP. All CT examinations were performed with a 64-slice CT tomography unit GE Light Speed Pro (GE Healthcare, Milwaukee, WI, USA), with and without intravenous injection of 100 mL water-soluble iodine contrast medium (270–320 mg/mL) in parenchymal phase approximately 40 s after intravenous contrast material administration. The slice thickness was 2.5 mm. All images were analyzed at a window level of 40 Hounsfield units (HU) and window width of 300 HU. All abdominal CECT scans were done in the cranio-caudal direction, with patients lying in the supine position, and holding breath in deep inspiration. CECT scans were evaluated by at least two experienced abdominal radiologists.

The pancreas was identified based on the typical landmarks (splenic vein and superior mesenteric artery). The left part of the pancreas is separated from the right part at the isthmus of the pancreas.

Pancreatic necrosis was diagnosed when any part of pancreatic parenchyma demonstrates attenuation of less than 30 HU during the parenchymal phase (normal pancreatic parenchyma demonstrates maximum enhancement typically, 100–150 HU). The severity of necrotizing AP at imaging was determined based on the extent of parenchymal involvement by necrosis (i.e., <30%, 30–50% and >50%). The diagnosis of peripancreatic necrosis was diagnosed by the presence of increased attenuation in peripancreatic fat, linear stranding and fluid collections, visible among the peripancreatic fat.

### 2.3. Treatment

All patients were treated according to the standard treatment protocol based on current international guidelines [8,11,12]. If moderately severe or severe AP was predicted and nausea and vomiting were present, enteral nutrition was initiated within the first 24–48 h. Pain management was based on a stepwise pain management protocol of AP starting with nonsteroidal anti-inflammatory drugs (NSAIDs). Prophylactic antibiotics were only administered for severely ill patients treated in the ICU, otherwise, antibacterial treatment was started only if infected pancreatic necrosis was diagnosed or suspected. Minimally invasive step-up approach interventional management was used when conservative treatment of infected pancreatic/peripancreatic necrosis was unsuccessful [13,14]. Ultrasound- or CT-guided percutaneous catheter drainage was followed by a necrosectomy if the status of the patient did not improve. Routinely, no antibiotic prophylaxis was given. Patients with persistent organ failure were treated at the intensive care unit.

### 2.4. Statistical Analysis

Statistical analysis was performed using SPSS 24.0 for Mac (SPSS Inc., Chicago, IL, USA). The data are presented as mean ± standard deviation or median and range. For comparison between groups, the Mann–Whitney test or Student’s *t* test were employed where appropriate. *p* < 0.05 was considered statistically significant.

## 3. Results

Data of 83 patients (76% males (*n* = 63)) with a mean age of 53 ± 1.7 years was analyzed. The mean time from symptom presentation until hospitalization was 88 ± 30.33 h. The majority of the patients (84% (70)) had the first episode of AP, and in half of the patients, the etiology of disease was unknown (*n* = 42, 51%). The volume of pancreatic necrosis exceeded 50% in half of the patients (*n* = 42, 51%). Abundant peripancreatic infiltration was found in 32 cases (38.6%). Detailed clinical data analysis is presented in Table 1 and Table 2.

### 3.1. Clinical Outcome by the Extent of Pancreatic Necrosis

If a large-volume (>50%) pancreatic necrosis was identified, patients mostly were classified as having severe AP (*p* < 0.05). If the volume of necrosis was equal to or less than 50%, the majority of patients were classified as having moderately severe AP (*p* < 0.05). There were no patients diagnosed with mild AP in this group. MODS score on admission was also higher in patients with large volume pancreatic necrosis when compared to that of patients with less than 30% of pancreatic necrosis (*p* = 0.008).

In the group of patients with pancreatic necrosis exceeding 50%, ultrasound (US)-guided interventions (26 (62%)) and open surgical treatment (17 (74%)) were performed more often when compared to those in patients with lesser extent of pancreatic necrosis. Open necrosectomy was performed in 19 (23%) patients, retroperitoneoscopy—in four (5%) patients. Forty-one percent (41%) of patients with pancreatic necrosis exceeding 50% underwent operation. All retroperitoneoscopies were performed in patients with a large extent (>50%) of pancreatic necrosis. Infected pancreatic necrosis was diagnosed in 33 cases (40%). Among patients with pancreatic necrosis exceeding 50%, infected necrosis was diagnosed in half of the cases (22, 52%). Contrarily, in patients with lesser-extent (<30%) pancreatic necrosis, infection was diagnosed in five cases (23%, *p* = 0.021). In addition, higher incidence of renal (29%, *p* = 0.036) and respiratory (49%, *p* = 0.024) failures and multiple organ dysfunction syndrome (41%, *p* = 0.017) was observed in patients with pancreatic necrosis exceeding 50%. These patients were treated longer in intensive care units (ICU) and stayed longer in the hospital compared to patients having smaller amounts of pancreatic necrosis (<30%, *p* = 0.003 and 30–50%, *p* = 0.027). The highest mortality rate was also in patients with pancreatic necrosis exceeding 50%. Detailed data is presented in Table 3.

### 3.2. Clinical Outcome by the Location of Pancreatic Necrosis

Patients with necrosis involving the whole pancreas were more often diagnosed with severe AP (*p* > 0.05). In this group of patients, the rate of complications, the rate of infected pancreatic necrosis and the need for surgical interventions were higher. These patients also stayed longer in the hospital and were treated longer in the ICU. The highest in-hospital mortality rate was also in the group of patients with whole-pancreatic necrosis (Table 4).

## 4. Discussion

Acute pancreatitis is one of the most common diseases of the gastrointestinal tract and in many cases is associated with a wide range of complications and potentially lethal outcomes. There are several classifications based on AP severity. Revised Atlanta Classification or Determinant-based classification is used in most of the centers worldwide. According to its severity, AP has different prognoses. Most of the published studies demonstrate an association between severe AP and worse prognosis and higher risk for an intervention [15,16]. During the course of the disease, pancreatic necrosis develops in a significant part of patients with AP. However, the significance of extent and location of pancreatic necrosis remains controversial.

The results of the present study showed that the clinical course and outcomes were worse in the cases of pancreatic necrosis exceeding 50%. Almost in half of the cases with >50% of pancreatic necrosis (41%), multiple organ dysfunction syndrome developed. These patients also had a higher rate of MODS score on admission. Their need for a minimally invasive step-up approach as well as open surgical treatment was higher. The more of the pancreas was necrotized, the bigger the risk of complications. Data from the literature show that patients diagnosed with MODS are reported to have worse outcomes [17]. Our findings that large extent (>50%) of necrosis and necrosis of the entire pancreas was associated with higher incidences of MODS confirm this statement. Our results show that these patients stayed longer in the hospital and needed more surgical interventions. In addition, with the increase of the extent of necrosis, the mortality rate also increased. Therefore, timely diagnosis of pancreatic necrosis and evaluation of its extent is very important. This statement is supported by a recent retrospective study from Spain showing that the volume of pancreatic necrosis significantly correlated with complications of AP (organ failure, multiple organ failure, infection, need of treatment, hospitalization at ICU) and might be an important radiological biomarker [18]. The significance of the necrosis volume for patients with acute necrotizing pancreatitis was also shown in a study from China. The authors found that there is a significant association between the volume of pancreatic necrosis together with mean CT density and complications (organ failure, need for interventions) [19]. However, it is still not clear if pancreatic necrosis is a prerequisite for the development of MODS in patients with AP. Researchers from the University of Edinburgh, United Kingdom, found an association between pancreatic necrosis and MODS in more than half of analyzed cases (58%; 95% confidence intervals (CI), 52.1–63.8%). They concluded that there is an association between pancreatic necrosis and development of MODS. However, it is still not clear if pancreatic necrosis is the cause of MODS [20].

Computed tomography is the “gold” standard when diagnosing pancreatic necrosis and other complications of AP. Therefore, the role of CT in predicting AP severity and the course of disease is significant [21]. Timing of initial and repeated CT in AP management is still a topic of ongoing discussions [17,21]. According to IAP/APA guidelines, initial CT should be assessed at least 72–96 h after symptom presentation [6]. However, there are some studies proving that early (within 72 h) CT scans can predict complications and improve management of AP [17]. Follow-up CT scans are proven to be a tool for prediction of local and systemic complications of AP. In severe AP, it is recommended to repeat the CT scan after 7–10 days after the initial CT [11]. Moreover, recent AP management guidelines suggest that in most cases of mild AP, computed tomography is not mandatory [11]. Contrarily, CECT is the main tool for diagnosis of necrosis at its extent and volume in patients with necrotizing AP.

Management of AP, especially in case of pancreatic necrosis, is challenging. Moreover, it has changed during the last 20 years. Patients in the present study were treated according to the guidelines of that period of time. IAP/APA evidence-based guidelines in 2013 suggested to start the treatment initially with restricting oral feeding and fluid resuscitation. In most of the cases, as recommended, intravenous antibiotic prophylaxis was not given. If infected pancreatic necrosis is diagnosed, the guidelines recommended starting with minimally invasive treatment [17,21]. However, interventional treatment should be avoided in the first weeks and postponed, if clinical status allows, for at least four weeks. If the condition of the patient deteriorates, the treatment should be started early [12]. As the optimal strategy, the first step should be percutaneous or endoscopic drainage [8]. However, which method to choose is still an open question [13]. In our study, if a minimally invasive approach was needed, it was started with percutaneous puncture und eventually drainage. If necessary, open necrosectomy remains a following step in the treatment of necrotizing AP. Current data shows that a step-up approach is associated with lower major complications and mortality rate [14]. Nonetheless, it is agreed that in some selected cases, open necrosectomy might still be the treatment of choice [13]. Our data shows that patients with pancreatic necrosis exceeding >50% are more likely to undergo surgical intervention. In addition, these patients stay longer in the hospital and in the ICU.

## 5. Conclusions

In conclusion, we showed that in patients with pancreatic necrosis exceeding 50%, the clinical course and outcomes were worse. These patients most often developed severe AP and spent more time in the hospital and ICU. These patients also more often needed surgical interventions, and the treatment was more complex. Therefore, timely diagnosis of pancreatic necrosis and evaluation of its volume and extent is highly important in the management of patients with acute necrotizing pancreatitis.

## 6. Limitations

Firstly, the sample size was too small to find more significant differences between groups. Secondly, some patients might not have been included in the study because of database imperfection. Finally, some patients might have been eliminated from the study after radiological evaluation because it was difficult to segregate peripancreatic and pancreatic necrosis, and significance of extrapancreatic necrosis was not analyzed.

## Figures and Tables

**Table 1 medicina-58-00645-t001:** Clinical data by the extent of pancreatic necrosis.

	Extent of Pancreatic Necrosis		
	<30%	30–50%	>50%
Patients, *n* (female/male ratio)	22 (5/17)	19 (5/14)	42 (10/32)
Mean age, years	49.3 ± 3.7	53.2 ± 3.4	55.1 ± 2.2
Time from symptom presentation until hospitalization, hours	138.2 ± 96.8	39.8 ± 8.2	83.5 ± 32.4
Recurrent AP, *n* (%)	4 (18.2)	5 (26.3)	4 (9.5)
Etiology, *n* (%)			
Alcohol-induced	5 (22.7)	9 (47.4)	12 (28.6)
Biliary	3 (13.6)	3 (15.8)	9 (21.4)
Idiopathic	14 (63.6)	7 (36.8)	21 (50.0)
Peripancreatic infiltration	9 (28.1)	8 (25)	15 (46.9)

AP—acute pancreatitis.

**Table 2 medicina-58-00645-t002:** Clinical data by the location of pancreatic necrosis.

	Location of Pancreatic Necrosis		
	Left Side	Right Side	Entire Pancreas
Patients, *n* (female/male ratio)	23 (7/16)	24 (16/18)	36 (7/29)
Mean age, years	49.5 ± 3.5	54.5 ± 3.1	54.5 ± 2.4
Time from symptom presentation until hospitalization, hours	35.9 ± 8.6	141.2 ± 88.6	85.8 ± 37.4
Recurrent AP, *n* (%)	4 (17.4)	5 (20.8)	4 (11.1)
Etiology, *n* (%)			
Alcohol-induced	3 (13.0)	9 (37.5)	14 (38.9)
Biliary	3 (13.0)	5 (20.8)	7 (19.4)
Idiopathic	17 (73.9)	10 (41.7)	15 (41.7)
Peripancreatic infiltration	12 (23.5)	8 (25)	12 (37.5)

AP—acute pancreatitis.

**Table 3 medicina-58-00645-t003:** Clinical outcome by the extent of pancreatic necrosis.

Extent of Pancreatic Necrosis			
	<30%	30–50%	>50%
Patients, *n* (%)	22 (26.5)	19 (22.9)	42 (50.6)
Revised Atlanta Classification			
Moderately severe AP, *n* (%)	18 (81.8) *	16 (84.2) *	24 (57.1) *
Severe AP, *n* (%)	4 (18.2) *	3 (15.8) *	18 (42.9) *
Determinant-based classification			
Moderate AP, *n* (%)	16 (72.7) *	14 (73.7) *	15 (35.7) *
Severe AP, *n* (%)	3 (13.6)	2 (10.5)	14 (33.3)
Critical AP, *n* (%)	3 (13.6)	3 (15.8)	13 (31)
MODS score on admission (range)	1 (0–4) *	2 (0–8)	2 (0–6) *
Interventions, *n* (%)			
US intervention	7 (31.8) *	7 (26.8)	26 (61.9) *
Operation	3 (13.0)	3 (15.8) *	17 (73.9) *
Complications, *n* (%)			
Infected necrosis	5 (22.7) *	6 (33.6)	22 (52.4) *
Sepsis	3 (13.6)	3 (15.8)	15 (35.7)
Renal failure	4 (18.2)	1 (5.3) *	12 (29) *
Respiratory failure	4 (18.2) *	3 (15.8) *	19 (49.2) *
Heart failure	2 (9.1)	1 (5.3)	10 (23.8)
Multiple organ dysfunction syndrome, number of patients	4 (18.2)	2 (10.5) *	17 (40.5)
Mean hospital stay, days (range)	17.9 ± 2.6 (3–54) *	23.7 ± 4.6 (5–83) *	49.2 ± 7.3 (6–195)
Mean days at ICU, days	4.1 ± 2.2 (0–35) *	1.8 ± 1.2 (0–22) *	12.8 ± 2.9 (0–59) *
In-hospital mortality	3 (13.6)	2 (10.5)	13 (31)

* *p* < 0.05, AP—acute pancreatitis, MODS—multiple organ dysfunction score, CRP—C-reactive protein, WBC—white blood cell, ICU—intensive care unit, US—ultrasound.

**Table 4 medicina-58-00645-t004:** Clinical outcome by the location of pancreatic necrosis.

	Location of Pancreatic Necrosis		
	Left Part	Right Part	Entire Pancreas
Number of patients, *n* (%)	24 (28.9)	23 (27.7)	36 (43.4)
Revised Atlanta Classification			
Moderately severe AP, *n* (%)	19 (79.2)	16 (69.6)	23 (63.9)
Severe AP, *n* (%)	5 (20.8)	7 (30.4)	13 (36.1)
Determinant-based classification			
Moderate AP, *n* (%)	18 (75)	13 (56.5)	14 (38.9)
Severe AP, *n* (%)	2 (8.3)	5 (21.7)	12 (33.3)
Critical AP, *n* (%)	4 (16.7)	5 (21.7)	10 (27.8)
MODS score on admission (range)	1 (0–4)	2 (0–8)	2 (0–6)
Interventions, *n* (%)			
US intervention	8 (33.3)	10 (48.5)	22 (61.1)
Operation	4 (16.7)	6 (26.1)	13 (36.1)
Complications, *n* (%)			
Infected necrosis	4 (16.7) *	10 (48.5)	19 (52.8) *
Sepsis	5 (20.8)	6 (26.1)	10 (27.8)
Renal failure	1 (4.2) *	5 (21.7)	11 (30.6) *
Respiratory failure	5 (20.8)	7 (30.4)	14 (38.9)
Heart failure	4 (16.7)	2 (8.7)	7 (19.4)
Multiple organ dysfunction syndrome, number of patients	4 (16.7)	6 (26.1)	13 (39.4)
Mean hospital stay, days (range)	19.1 ± 2.9 * (5–66)	30.1 ± 5.7 * (3–104)	48.8 ± 8.2 * (6–195)
Mean days at ICU, days	2.3 ± 1.3 * (5–66)	7.0 ± 2.6 (3–104)	12.4 ± 3.3 * (0–59)
In-hospital mortality, *n* (%)	4 (16.7)	4 (17.4)	10 (27.8)

* *p* < 0.05, AP—acute pancreatitis, MODS—multiple organ dysfunction score, ICU—intensive care unit, US—ultrasound.

## Data Availability

Not applicable.

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
