# Peer review of "Volume, but Not the Location of Necrosis, Is Associated with Worse Outcomes in Acute Pancreatitis: A Prospective Study"

_medicina, 2022, doi:10.3390/medicina58050645_

Round 1

Reviewer 1 Report

The manuscript of Dekeryte and co-workers is extensive and interesting. The overall composition of the manuscript is good, is written in a clear and understanding manner with brief introduction, however it does contain a few typos. The quality of the tabeles is also good. The paper is scientifically and methodologically accurate, nevertheless with some limitations. The major limitation of this study is the size of study group, especially when are divided for subgroups.  However, the authors are aware of the limitations, and confirmed it in separate section. Besides, the conclusions drawn are convincing.

Author Response

Thank you for your positive opinion about our manuscript. We appreciate your comments.  

Reviewer 2 Report

   In this manuscript, the authors presented that necrosis volume was associated with worse outcomes. However, I advise the authors to make some modifications in the manuscript.

Major points

#1.    In the materials methods section, the last paragraph of page two, the authors described about the extent of pancreatic necrosis. I request the authors to explain more detail. What means <30%? Less than 30% of the volume of hole pancreas? How did the authors calculate volume?

#2.    Table 1 seems not correct. The line is incorrect, and some overlapping characters. Please modify it.

#3.    I request the authors to simplify table 3 and table 4. There are too many items. Also, I request the authors to simplify description of the results section of “3.2. Clinical outcome by the location of pancreatic necrosis”. Please describe the results more simply.

Author Response

Comment #1:

In the materials methods section, the last paragraph of page two, the authors described about the extent of pancreatic necrosis. I request the authors to explain more detail. What means <30%? Less than 30% of the volume of whole pancreas? How did the authors calculate volume?

Reply #1:

Thank you for your question. In this study all the CECT scans were evaluated by 2 radiologists. The extent of pancreatic necrosis was calculated firstly excluding peripancreatic necrosis. Then the images were reviewed in all 3 planes (axial, coronal and sagittal) and necrosis extent of the whole pancreas was assigned to one of 3 groups (<30%, 30-50% or >50%). All unclear cases were discussed among two radiologists.

Comment #2:

Table 1 seems not correct. The line is incorrect, and some overlapping characters. Please modify it.

Reply #2:

Thank you for this remark. We have corrected the table.

Comment #3:

I request the authors to simplify table 3 and table 4. There are too many items. Also, I request the authors to simplify description of the results section of “3.2. Clinical outcome by the location of pancreatic necrosis”. Please describe the results more simply.

Reply #3:

Less important data from Table 3 and Table 4 was removed. Section 3.2 was rewritten to make it more simple and easier to understand.

Round 2

Reviewer 2 Report

In this manuscript, the authors presented that necrosis volume was associated with worse outcomes. I believe this paper has some clinical implications, and I think this revised paper is worth publication.